# Solid-State Electrolyte for Lithium-Air Batteries: A Review

**DOI:** 10.3390/polym15112469

**Published:** 2023-05-26

**Authors:** Qiancheng Zhu, Jie Ma, Shujian Li, Deyu Mao

**Affiliations:** School of Mechanical and Automotive Engineering, Guangxi University of Science and Technology, Liuzhou 545006, China; lz2392737170@163.com (J.M.); lsj3207043815@163.com (S.L.)

**Keywords:** solid-state electrolyte, lithium-air batteries, inorganic solid electrolyte, polymeric solid electrolyte, composite electrolyte

## Abstract

Traditional lithium–air batteries (LABs) have been seriously affected by cycle performance and safety issues due to many problems such as the volatility and leakage of liquid organic electrolyte, the generation of interface byproducts, and short circuits caused by the penetration of anode lithium dendrite, which has hindered its commercial application and development. In recent years, the emergence of solid-state electrolytes (SSEs) for LABs well alleviated the above problems. SSEs can prevent moisture, oxygen, and other contaminants from reaching the lithium metal anode, and their inherent performance can solve the generation of lithium dendrites, making them potential candidates for the development of high energy density and safety LABs. This paper mainly reviews the research progress of SSEs for LABs, the challenges and opportunities for synthesis and characterization, and future strategies are addressed.

## 1. Introduction

The first generation of lithium ion solid-state electrolytes dates back to the 1830s when Faraday discovered that heated Ag_2_S and PbF_2_ had anionic conduction properties, but they did not develop rapidly because of they have low ionic conductivity and high interfacial impedance at room temperature, and susceptibility to short circuit due to dendrite penetration [1]. It was not until the 1960s that β-Al_2_O_3_ was discovered to have two-dimensional sodium ion conduction properties and was subsequently applied to high-temperature sodium-sulfur batteries [2,3]. Therefore, the 1960s is considered the beginning of the development of solid-state electrolytes, which began to be used in batteries. In the following decade, Ag_3_SI solid-state lithium-ion conductor materials were successfully used for energy storage; solid-state electrolytes are increasingly used in practical applications [4], and in 1973, the PEO polymer was discovered to have the ability to conduct lithium ions, thus expanding the scope of solid-state ionics from inorganic materials to polymers [5], and since then, lithium–ion polymer conductors have emerged and been used in all-solid-state polymer lithium–ion batteries. In 1992, Oak Ridge National Laboratory successfully prepared LiPON thin film electrolyte material, which played a key role in improving the performance of thin film lithium batteries [6]. Since then, many types of inorganic solid-state electrolyte materials have emerged, including chalcogenide, sodium supersonic conductor (NASICON), garnet, sulfide, etc. [7,8,9,10]. Until the early 21st century, solid-state electrolytes began to be combined with gaseous and liquid cathodes for lithium–ion batteries, such as solid-state lithium–air batteries, lithium–sulfur batteries, lithium–bromine batteries, etc. [11,12,13].

With the development of society, the energy density of current lithium–ion batteries is becoming more and more difficult to meet the demand. As a result, research and development of batteries with high energy density has been started, such as LABs, zinc–oxygen batteries, lithium–sulfur batteries, etc. [14,15,16]. The energy density of traditional lithium–ion batteries (less than 350 Wh kg^−1^) are increasingly inadequate for current energy storage devices, and there is an urgent need to find environmentally friendly energy storage devices with an energy density comparable to fossil fuels [17,18]. LABs have a high theoretical energy density (3505 Wh kg^−1^) [19], about 10 times the energy density of commercial lithium–ion batteries [20,21]. Depending on the reaction gas, the Li–CO_2_ battery system has recently developed with both high theoretical specific energy (1876 Wh kg^−1^) [22,23] and the ability to fix/convert CO_2_, which makes it a promising energy storage system [24].

Typical solid-state LABs consist of a porous cathode, an electrolyte, and a lithium anode, as shown in Figure 1. Electrolyte is the key component in the battery, which not only provides efficient ion transport capacity, but also blocks the electron transfer between cathode and anode, its performance directly influences the battery cycle life and energy density. Traditional liquid organic electrolytes are not suitable for future commercial LABs, due to following reasons [25,26,27,28,29]:The volatile and leaky problems seriously affect the stability of the battery system.Lithium dendrite growth may puncture the electrolyte diaphragm leading to cell short circuit.The reaction path may be changed by the byproducts induced from electrolyte decomposition.Water, oxygen, and other components in the ambient air inevitably pass through the electrolyte diaphragm and react with the lithium in the anode causing corrosion of the lithium, deteriorate battery performance.

On the contrary, solid-state LABs have the following advantages [30,31,32]:
A solid-state electrolyte has sufficient mechanical strength and superior electrochemical stability to be compatible with the high energy density lithium metal anode and high potential cathode contacts to achieve safety and high energy density.Solid and gel electrolytes are simple to prepare and easy to shape and manufacture in large quantities, reducing the difficulty of designing battery management systems.Due to the absence of liquid media, the recovery process is less difficult and further reduces costs.Solid electrolytes have higher thermal stability and safety than liquid electrolytes. Encapsulated cool systems are not necessary, reducing the cost of accessories.

In general, solid electrolyte instead of liquid electrolyte is the inevitable trend of future battery development. However, in practice, the following problems still exist with solid electrolytes [33,34,35].

SSEs have low ionic conductivity, especially at low temperatures.High interface impedance of electrode–electrolyte solid interface.Poor electrochemical compatibility with lithium metal cathodes.The weak physical stability of the electrode, resulting in large interfacial stress changes.

These problems seriously affect the stability, actual capacity, and life cycle of solid-state LABs. In the past decade, researchers focus on SSE, cathode materials, and anode materials in response to these problems. As an important part of the battery, SSE connects the cathode and anode, the forming interface is particularly important to determine the battery’s cycle stability and life. Therefore, optimization and improvement of SSE are key factors to promote the development of LABs.

SSEs can be divided into three categories [36]: inorganic solid electrolytes, polymer solid electrolytes, and composite solid electrolytes. This review introduced these SSEs in the following section, and the direction of solid-state LABs was prospected.

## 2. Solid-State Electrolytes

### 2.1. Inorganic SSEs 

Inorganic SSEs have a wide range of applications due to their high mechanical strength, wide potential energy range, and safety [37,38]. Inorganic SSEs include oxide-based SSEs and sulfide-based SSEs [39]. Sulfide-based SSEs have realized a preliminary application in Li–ion batteries, however, this category of SSEs may not be suitable for LABs. Owing to the moisture absorption ability [40], most sulfide-based SSEs are unstable in the atmosphere, and the toxic H_2_S resulting from the reaction will cause environmental problems, hindering their application [41,42]. In addition, the electrochemical stability of sulfide-based SSEs is poor when paired with a lithium anode or a high-voltage cathode. S^2-^ anions may undergo redox reactions, resulting in a narrow voltage window [43]. Compared to sulfide-based SSEs, oxide-based SSEs exhibit better mechanical stability and higher ionic conductivity. Oxide-based SSEs mainly contain [44,45,46]: garnet Li_7_La_3_Zr_2_O_12_ (LLZO and LLZTO), perovskite (LLTO), zeolite, and NASICON (LATP and LAGP).

#### 2.1.1. Garnet Based SSEs

Garnet is an attractive and promising solid electrolyte that has made significant progress in battery performance, achieving high energy density, high ionic conductivity, high stability to lithium metal, and a wide potential window [47].

Sun et al. [27] prepared a lamellar ceramic (LLZTO) solid electrolyte using a hot-pressure sintering technique with a density of 99.6% and a high ionic conductivity of 1.6 × 10^−3^ S cm^−1^ at room temperature, and used the LLZTO solid electrolyte cathode at 0.2 μm thicknesses as an ion-conducting framework with PI: LiTFSI combination (as shown in Figure 2a), with a high coulombic efficiency of 97.1% and good cycling performance in a lithium–air battery (SSLAB) at 200 °C. To reduce the operating temperature of the cells, the researchers replaced PI: LTFSI with PPC: LTFSI and LLZTO and recycled them at 80 °C. SEM images of the air cathode change during circulation. Compared to the pristine cathode (Figure 2b), rod-like particles were observed after full discharge (Figure 2c), indicating rapid growth and fusion of discharge products to form agglomerated particles. During charging, the large/condensed particles observed in Figure 2c contracted significantly to the compact particles observed in Figure 2d. After the fifth charge and discharge, similarities were observed between Figure 2e,c and between Figure 2f,d indicating repeated formation and decomposition of discharge products.

#### 2.1.2. Perovskite Based SSEs

High ionic conductivity, a wide electrochemical window, and high stability to lithium metal, perovskite electrolytes are promising solid electrolytes for batteries [48].

Le et al. [49] prepared an Al-doped LLTO solid-state electrolyte (A-LLTO) with a flexible chalcogenide structure using a citric acid-gel method. At a current density of 0.05 mA cm^−2^, the solid-state cell designed by this electrolyte operated stably in a pure O_2_ atmosphere from 25 °C to 100 °C and, respectively, provided a first discharge capacity of 796 mAh g^−1^ to 4035 mAh g^−1^. At 50 °C, the cell sustained 132 cycles at a high current density of 0.3 mA cm^−2^ and a limited capacity mode of 500 mAh g^−1^. This indicates a successful first step toward achieving a high-energy, high-cyclability, and high-safety lithium battery.

#### 2.1.3. Zeolite Based SSEs

Zeolite has a high specific surface area and is widely used in molecular adsorption, gas separation and catalyst carriers. It has good compatibility with lithium and air and exhibits superior electrochemical stability and high ionic conductivity in the ambient air of integrated solid-state LABs [50,51].

Chi et al. [52] used a stable and flexible zeolite electrolyte in solid-state LABs that remained constant at room temperature for one year and had high water stability. LiX zeolite pellets LiXZM were prepared by conventional plate pressing. LiXZM’s tightly arranged structure ensures smooth migration of ions between the crystals and the thickness of LiXZM is only about 5 μm, it achieves a charge/discharge cycle life of 149 cycles in LABs, which greatly improves its cycle life in the air. This Li^+^ exchange zeolite X (LiX) membrane SSE has become one of the most attractive materials for LABs. 

Chen et al. [53] has added a novel additive-ZSM-5 nano zeolite to electrolytes to improve battery life cycle. The addition of this zeolite molecular scavenger removes electrolyte breakdown products and provides superior performance compared to conventional organic additives. The capacity retention rate increased from 40% to 62% after 480 cycles. This is due to the enhanced stability of the interface layer, which improves cell life cycles.

#### 2.1.4. NASICON SSEs

NASICON-type ceramic LATP solid-state electrolytes have attracted attention for their air stability and rapid Li^+^ conductivity [54,55]. 

Na et al. [36] prepared LATP SSEs with smooth surfaces by polishing. LATP SSEs have a covalent network that TiO_6_ octahedra and PO_4_ tetrahedra sharing corners (Figure 3a). The covalent network provides a free channel for Li^+^ transport, thus improving the conductivity of Li^+^. Since the smaller Al^3+^ can replace the larger Ti^4+^ ions, the partial replacement of Ti^4+^ by Al^3+^ in the LATP crystal structure leads to a smaller unit cell size and a denser material. The dense surface can effectively inhibit the generation of lithium dendrites. Li–CO_2_ batteries with LATP solid-state electrolytes reach a maximum capacity of 5255 mAh g^−1^ at a current density of 60 mA g^−1^ (Figure 3b) and exhibited good cyclability (50 cycles) with a cutoff capacity of 600 mAh g^−1^ (Figure 3c).

LAGP SSEs exhibit a good relative stability both in the external environment and in contact with lithium metal, which have a wide electrochemical window (up to 6 V) in humid environments. LAGP ceramic electrolytes have high ionic conductivity and electrochemical stability, showing great potential for the development of high energy density all-solid-state LABs [57,58]. Although the LAGP solid electrolyte is stable in air, it is not sufficient to inhibit the growth of Li dendrites; short-circuiting still occurs even at low current densities [59]. Researchers found that this phenomenon is caused by the rough surface of the microstructure of the LAGP fabricated by conventional methods, and the location where Li dendrites occur is due to the sharp surface or edge. Subsequently, Wang et al. [60] prepared ultra-fine surfaces nanoscale LAGP solid-state electrolytes (UFSLAGP) using the nano-polishing technique and formed a dense and uniform electrolyte surface structure using the polishing technique. The polished UFSLAGP resulted in a uniform lithium deposit and exhibited a good charge/discharge cycling performance at a current density of 400 mA g^−1^ at room temperature, which effectively inhibits the growth of Li dendrites.

Zhang et al. [56] used the spin-coating cosintering method to add an artificial LAGP thin sheet glass protective layer to the ceramic LAGP surface. Forming in the surface amorphous electrolyte LAGP @ glass, it is continuous, thin, flat, dense (Figure 3d), and has no visible grains or grain boundaries on the cross-section (Figure 3e); it can inhibit lithium dendrites. Li/Li symmetric cells exhibited a good interfacial stability at a constant current of 0.1 mA cm^−2^ (Figure 3f). In LFP/LAGP@glass/Li battery, the initial capacity was 152.2 mAh g^−1^ and the capacity remained 93.6% after 120 discharge cycles (Figure 3g). Compared to traditional LiPON protection layer, the surface amorphous LAGP spin-coating cosintering process has the advantages of simplicity and economy, and good cycling stability.

In conclusion, in inorganic oxide solid electrolytes, NASICON, garnet, and perovskite solid electrolytes have high ionic conductivity, but the instability of some NASICON and garnet solid electrolytes to air and lithium metal requires careful consideration. Zeolite has good ionic conductivity and chemical stability, which provides a broad prospect for the preparation of solid-state lithium anodes.

### 2.2. Polymeric Solid Electrolytes

Solid polymer electrolytes (SPEs) have the following advantages over liquid electrolytes [61,62]:Improving safety due to its non-flammability and stability due to its high temperature resistance.Increasing energy density due to its excellent stability to lithium metal anodes.Removal of the polymer separator improves design flexibility and reduces manufacturing costs, among other advantages.

The disadvantage of polymer electrolytes at room temperature is the relatively low conductivity of lithium ions [63,64]. Researchers tried to improve this conductivity by blending, modifying, and preparing PEO derivatives in different methods [65]. Polymer electrolytes are usually composed of polymeric substrates such as [66]: polyethylene oxide (PEO), polyvinylidene fluoride (PVDF), polyvinylidene fluoride-hexafluoropropylene (PVDF-HFP), polyacrylonitrile (PAN), and poly methacrylate (PMMA), as well as some lithium salts (LiO_4_, LiFSI, Asf_6_), with low mass and low electrolytes.

#### 2.2.1. Polyethylene Oxide (PEO)

PEO-based polymers have the advantages of simple preparation, good mechanical elasticity, low interfacial resistance, and good stability with lithium metal. However, poor mechanical strength and low ionic conductivity at room temperature (10^−8^~10^−7^ S cm^−1^) of PEO-based all-solid electrolytes are due to the restricted chain motion [67]. Therefore, the use of composite electrolytes can compensate for their low mechanical strength and low ionic conductivity.

Su et al. [68] designed an all-solid lithium metal cell with a flexible PEO-LSPSCl-LiTFSI composite electrolyte with a capacity of 414 mAh g^−1^, a current density of 0.1 A g^−1^, a capacity retention rate of 97.8%, and an initial coulombic efficiency of up to 94% after 94 times cycle. The electrolyte has the better mechanical strength to inhibit lithium dendrite growth, thus improving stability of the Li/S-CPE interfacial stability.

#### 2.2.2. Polymethyl Methacrylate (PMMA)

PMMA-based polymer electrolytes are based on methyl methacrylate (MMA) or methyl propionate (MA). These polymer electrolytes typically have a wide window of electrochemical stability (>4.5 V), a high room temperature ionic conductivity (>10^−3^ S cm^−1^), and good compatibility with positive and negative electrodes. However, it has poor mechanical properties and usually needs to be used in combination with other substrate materials to utilize PMMA.

Kufian et al. [69] prepared a PMMA-based gel polymer electrolyte (GPE), where the gel polymer electrolyte used LiTFSI salt as the lithium–ion donor and TEGDME as the solvent. When the content of LiTFSI in the electrolyte increased from a certain range, the ionic conductivity of this gel polymer electrolyte also increases. The inverse convolution of 770~720 wave numbers was performed using FTIR spectroscopy (Figure 4a); when the LiTFSI content reaches 25 wt%, the conductivity reaches its highest, with the percentage of free ions reaching 54%. The highest ionic conductivity enhances the stability of the electrode/electrolyte interface; therefore, this PMMA-based polymer gel electrolyte has excellent ionic conductivity and good interfacial stability at room temperature for lithium–oxygen batteries.

#### 2.2.3. Polyacrylonitrile (PAN)

Polyacrylonitrile-based polymer electrolyte is one of the earliest polymer electrolytes. PAN polymers have advantages of chemical stability, non-flammability, thermal stability, cost, and price advantages [70,71]. In PAN, a pair of electrons on the N atom can interact with Li^+^ and increase the dissociation of the lithium salt, thus increasing the concentration of Li^+^ [72]. However, the PAN substrate is fragile, and therefore is not normally used as a substrate for polymer electrolytes alone. To compensate for this deficiency, ideal polymer electrolytes can be prepared by grafting, copolymerization, or blending with other polymer monomers with good mechanical properties. 

Tran et al. [63] prepared PAN/PVA blends, LATP, LiTFSI, and SN all-solid composite polymer electrolyte (PVAN50-20%LATP-10%SN) using solution fabrication techniques, which has a high ionic conductivity (1.13 × 10^−4^ S cm^−1^), a good thermal stability, a high tensile strength, and a smooth uniform surface that inhibits the growth of lithium dendrites (Figure 4b). At room temperature, ASSLMBs had a capacity retention rate of 98.3% at 0.1 C and the coulombic efficiency remained constant at 99.0% (Figure 4c).

#### 2.2.4. Polyvinylidene Fluoride (PVDF) 

PVDF-based polymer electrolyte has a high conductivity and a good electrochemical stability; PVDF chains contain the strong electron absorption group CF, so the polymeric solid electrolyte has a wide electrochemical stability window of 4.5 V or more [73], but its mechanical strength is still insufficient for practical applications.

Zhang et al. [74] discovered a water-soluble polymer hydroxyethyl cellulose (HEC) with a good thermal stability and good electrochemical properties as a stabilizer, binder, emulsifier, suspending agent, and dispersant; with a wide viscosity range, its dense structure can avoid microshort circuit problems in lithium–ion batteries. PVDF combined with the advantages of HEC to prepare a PVDF/HEC/PVDF sandwich GPE combination (Figure 5a,b), and the two materials carried out to complement each other.

#### 2.2.5. Polyvinylidene Fluoride-Hexafluoropropylene (PVDF-HFP)

Due to the single structure of PVDF polymers, its crystallinity within the molecule is high; furthermore, the -F in the chain is easy to react with lithium metal using multiple charge/discharge cycles causing the hydrophobicity of the membrane to decrease, thus causing moisture in the air to enter and destroy the lithium negative interface. To solve the above problems, polyvinylidene fluoride-hexafluoropropylene (PVDF-HFP) with oxygen selectivity was experimentally studied. Oxygen selective membrane (OSM) has the absolute barrier to other gas components in the air, especially H_2_O.

Based on PVDF, Wen et al. [75] prepared a porous polyvinylidene fluoride-hexafluoropropylene (PVDF-HFP) oxygen-selective membrane using electrostatic spinning, and further infiltrated the PFPE into the PVDF-HFP fiber membrane, where PFPE fills 11.3% of the pore (Figure 5d,e), obtaining a PFPE@PVDF-HFP hydrophobic membrane that both guarantees the integrity of the fiber skeleton and serves as a hydrophobic barrier (Figure 5c). To highlight the hydrophobic properties of the hydrophobic membrane, the researchers selected different protective measures in ambient air (relative humidity ~50%) and observed the degree to which the crystallization of silica gels to test tubes changed after exposure to water (the crystalline color of cobalt chloride gradually changed from blue to light red after exposure), the hydrophobic film remained blue after 40 days of placement (Figure 5f,g). The lithium–air battery can be cycled stably for 620 h in ambient air at 50% relative humidity, and the discharge capacity of the battery under PFPE@PVDF-HFP protection at a constant current density of 100 mA g^−1^ is 6019 mAh g^−1^, which is a good improvement of the cycle life and the hydrophobic effect.

Ren et al. [76] used polyvinylidene fluoride-hexafluoropropylene (PVDF-HFP) as a backbone and UV in situ polymerization of trimethylpropane polyoxyethylene ester triacrylate (TEMPT) within the PVDF-HFP network, followed by the addition of OAC redox medium to obtain a DPGE-OAC electrolyte. The electrolyte can be charged and discharged for 200 cycles at a current density of 0.4 mAh cm^−2^ (Figure 5i), improving the cycle life and stability of the battery.

In conclusion, although crystallization results in low Li^+^ conductivity in polymeric solid electrolytes, the decomposition of polymeric substrates can limit their side reactions in the open environment. The polymeric solid electrolyte has excellent tolerance and a simple preparation process, so it is ideal to develop flexible solid-state LABs.

### 2.3. Organic-Inorganic Composite Solid Electrolytes

Organic-inorganic composite solid electrolytes are often a mixture of organic and inorganic phases, which achieves functional complementarity of each component. Polymeric SSEs have good flexibility and interfacial stability. However, the ionic conductivity of polymeric solid electrolytes is very low at room temperature, and most all-solid-state batteries based on polymeric solid electrolytes can only operate at high temperatures [77]. Inorganic SSEs have high strength and high conductivity, but high hardness, poor processing performance, and high interfacial impedance [78]. Therefore, to obtain SSE with suitable properties, a composite of organic and inorganic has good features that are compatible with the flexibility and interfacial stability of polymers and include the advantages of high ionic conductivity and high strength of inorganic materials. Therefore, the study of composite SSEs has become an important method to improve the performance.

Xu et al. [79] prepared an electrolyte that immobilizes and retains a liquid electrolyte in a gel polymer substrate (PI@GPE). It has a uniform and dense gel-like structure (Figure 6a) that can effectively inhibit the decomposition of the liquid electrolyte, resulting in a high ionic conductivity of 0.44 mS cm^−1^ and Li^+^ mobility number of 0.596. It exhibits the superior interfacial compatibility in contact with lithium metal, thus improving interfacial stability. With the better hydrophobicity in the air, water vapor transport is inhibited in the surrounding air to protect lithium metal. The PI@GPE electrolyte in a lithium–oxygen cell exhibited stable performance at 0.1 mA cm^−2^ and 0.25 mAh cm^−2^ cycling capacity of 366 cycles (Figure 6b). 

Song et al. [80] modified garnet Li_7_La_3_Zr_2_O_12_ (LLZO) as an active filler to compound composite solid electrolyte (3D-CPE) with polymer polyethylene oxide (PEO) (Figure 6d). The modified LLZO filler network is well connected, and the grains have several particulate microporous structures (Figure 6e), which facilitates the incorporation of PEO polymer. In Figure 6g, cross-sectional images show that the thickness of 3D-CPE is about 200 μm, and the 2 μm SEM images further confirm that the structure of 3D-CPE is dense and that the 3D LLZO framework maintains the original structure. The 3D-CPE composite solid-state electrolyte can make the formation of more conductive amorphous phases from the less conductive crystalline phases and provide ion transport channels along the polymer/filler interface and bulk phase, thus exhibiting extremely high ionic conductivity (~10^−4^ S cm^−1^). In addition, it can be cycled for 50 cycles at a limited capacity of 300 mAh g^−1^ in LABs, exhibiting excellent cyclability and superior stability.

Zhao et al. [81] proposed a 3D porous garnet/gel polymer hybrid solid electrolyte (PSSE/GPE), the hybrid electrolyte synthesis process is shown in (Figure 7a). In this hybrid solid electrolyte, the 3D rigid skeleton microstructure of PSSE can suppress Li dendrites, and the 3D rigid skeleton of PSSE has conductivity that provides “hiking paths” for an additional supply of lithium ions. The continuous GPE in PSSE provides high ionic conductivity can be used as a “sailing path” for a large amount of Li–ion transport (Figure 7c). Additionally, the mixture of PSSE and GPE eventually achieves a higher ionic conductivity (1.06 × 10^−3^ S cm^−1^). In addition, PSSE/GPE has a good ability to isolate moisture, oxygen, and carbon dioxide from the air, and has good electrochemical stability with a lithium metal anode, enhancing the safety and cycling stability of Li–O_2_ batteries. The PSSE/GPE-based Li–O_2_ battery has a high cycle capacity (1250 mAh g^−1^) and a long charge-discharge cycle life (up to 194 weeks).

In recent years, solid lithium battery electrolytes with inorganic inert fillers and organic polymers have cycling performance and stability. The addition of inorganic inert fillers enhances amorphous regions and ion transport in solid polymer electrolytes, which is the most effective way to improve ionic conductivity, mechanical strength and expand the window of electrochemical stability [82]. Inorganic inert fillers include silicon dioxide (SiO_2_), magnesium oxide (MgO), alumina (Al_2_O_3_), titanium oxide (TiO_2_), etc. [83]. They are oxide ceramic fillers and do not involve Li^+^ transport [84,85]. The effect of inorganic inert filler on the ionic conductivity of solid composite electrolyte is shown in three aspects. First, it decreases crystallinity. Second, it improves the interfacial stability of solid electrolyte. Third, it increases the number of cation transfer.

Gulino et al. [86] prepared nanocomposite solid electrolytes by mixing LiBH_4_ with the inorganic inert material MgO. The addition of MgO increased the formation of the LiBH_4_ conductive interface; it consists of a core-shell model; it improves the conductivity of lithium ions. With a lithium–ion conductivity of 2.86 × 10^−4^ S cm^−1^ at 20 °C, which is about four orders of magnitude higher than that of pure LiBH_4_. The operating temperature of pure LiBH_4_ is 120 °C, the addition of MgO reduces the operating temperature of LiBH_4_ to 60 °C or even to room temperature. When the temperature was at 60 °C and the current density was 11.8 mA g^−1^, multiple charge/discharge cycles in the Li|CE53|TiS_2_ solid-state battery could form a stable SEI layer, which improved the charge/discharge cycle capability of the battery (65 cycles). After the battery rested for 4 h and continued to operate at room temperature for 30 cycles, with a specific capacity of about 50 mAh g^−1^. The innovative point of this experiment was the formation of a stable SEI layer by high-temperature charge/discharge cycles, which resulted in the ability to operate at room temperature.

Wang et al. [87] prepared the all-solid electrolyte PMMA/m-MgO using external chemical modification of the inorganic inert material MgO. Stable oxygen-containing functional groups were introduced in the magnesium oxide modification process, and reduced the ionic conductivity distance, thereby increasing the ionic conductivity of the MgO electrolyte and combining it with PMMA/m-MgO polymers to prepare an all-solid electrolyte. At room temperature, the ionic conductance of PMMA/m-MgO electrolyte was significantly higher than that of the corresponding PMMA/MgO electrolyte under the same conditions. The reason for this difference is that the modification of MgO improves ion mobility. In addition, as the temperature increases, polymer activity increases, and the ionic conductivity increases at higher temperatures, thus increasing the transport capacity of lithium–ion transport. After testing solid-state Li–O_2_ batteries, the researchers found that PMMA/m-MgO electrolyte cells have a higher capacity and better multiplication than PMMA/MgO electrolyte, with up to 52 charging/discharge cycles (Figure 8b).

Yi et al. [88] prepared PVDF-HFP/ PMMA-based composite gel polymers doped with spherical ZrO_2_ nanofillers by solution casting method (Figure 8d), where PVDF-HFP has good mechanical strength, good chemical stability, and excellent dielectric conductivity, which can promote ion dissociation. It is considered as the most promising gel electrolyte backbone. When it was blended with PMMA and ZrO_2_ nanofillers doping, the polymer film exhibited a smooth and dense morphology (Figure 8e), where ZrO_2_ was uniformly distributed in the polymer. The lithium–ion conductivity is up to 1.46 × 10^−3^ S cm^−1^, with a wide electrochemical window of about 4.65 V (vs. Li^+^/Li), and 6% ZrO_2_ nanofiller content electrolyte which has a good tensile strength (37.7 MPa) (Figure 8f). Good interfaces in contact with stable lithium metal anodes can effectively achieve uniform deposition of Li and thus effectively inhibit the growth of lithium crystals.

Colombo et al. [89] designed a polymer ceramic nanocomposite electrolyte (PEO-grafted nanofiber TiO_2_) (Figure 8g) in which the addition of TiO_2_ filler enhanced the mechanical properties, mechanical strength, and interfacial stability of the composite electrolyte. A large number of inorganic nanomaterials can inhibit the production of Li dendrites and prevent short circuits, thus improving the battery cycle life. When the TiO_2_ content reached 39%, the composite electrolyte exhibited good specific capacity and high coulomb efficiency in a charge and discharge cycle at 70 °C (Figure 8h).

In conclusion, the inorganic ceramic solid electrolyte has a poor electrolyte compatibility with the electrode but a better ionic conductivity. Monomeric polymer solid electrolyte has a lower ionic conductivity at room temperature but a good flexibility and stability. Establishing a composite electrolyte is an effective way to obtain an ideal solid electrolyte and to compensate for the shortage of various electrolytes.

## 3. Prospect

Solid-state LABs including Li–CO_2_ batteries are the most alternative next-generation energy sources, with the following performances: (1) Non-volatile and non-flammable. (2) Lithium dendrite growth is suppressed. (3) Side reactions induced by organic electrolyte are avoided. (4) Air components (H_2_O, O_2_, CO_2_, etc.) are prevented. However, the technology is still far from application, innovations are encouraged, including sintering additives adding, cold sintering process, interface modification, and block copolymer.

### 3.1. Sintering Additives Adding 

Ceramic solid-state electrolytes typically have high ionic conductivity and excellent electrochemical stability [90], but the preparation process is affected by high temperature sintering, during which lithium loss and two-phase formation occur [91]. Therefore, the development of low-temperature sintering technology is the key to the preparation of solid ceramic electrolytes. In recent years, the addition of sintering additives Li_2_B_4_O_7_, LiBF_4_, Li_2_O, Li_3_PO_4_, LiBO_2_, Li_3_H_2_O, and LiF to solid electrolyte were found to reduce the sintering temperature. Furthermore, the addition of sintering additives can reduce the liquid phase of electrolyte materials at lower temperatures, accelerating particle growth, reducing stress from grain growth, obtaining densely sintered lithium sheets [92], thus reducing loss of lithium and generation of second phase.

Dai et al. [93] added LiBF_4_ to LATP powder to reduce the sintering temperature thus reducing the lithium–ion loss. The highest ionic conductivity was achieved when the LiBF_4_ content was 3 wt% (Figure 9a); it greatly improved the sintering performance of LATP at 800 °C sintering temperature. LiBF_4_ formed a liquid phase at a high temperature to promote grain growth and formed channels that more suitable for lithium–ion migration, thus improving the lithium–ion conductivity. 

Bai et al. [94] used LiBO_2_ as a sintering aid, investigated the effect of different sintering temperatures and contents on the ionic conductivity of LATP. When the LiBO_2_ content was 1 wt% and the sintering temperature was 800 °C (Figure 9c,d). The ionic conductivity was 3.5 × 10^−4^ S cm^−1^ at room temperature. The addition of lithium oxide reduced the sintering temperature, reduced the formation of two phases and improved the ionic conductivity.

**Figure 9 polymers-15-02469-f009:**
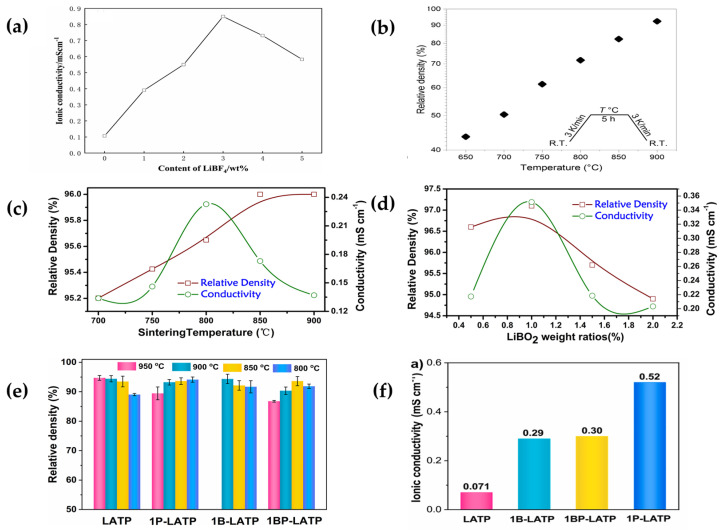
(**a**) The total ionic conductivity of ceramic LATP with different LiBF_4_ contents from 0 to 5 wt%. (Reprinted with permission from [93]; copyright 2021, Ceramics International). (**b**) Relative density of LATP particles sintered at different temperatures (reprinted with permission from [95]; copyright 2019, Solid State Ionics). (**c**) Sintering density and total ionic conductivity of LATP (T1-T5) ceramics sintered at different sintering temperatures. (**d**) Sintering density and total ionic conductivity of LATP (B1–B4) ceramics sintered at 800 °C with different LiBO_2_ contents (0.5–2.0 wt%) (reprinted with permission from [94]; copyright 2018, Ceramics International). (**e**) Relative densities of pure LATP, 1P-LATP, 1B-LATP, and 1BP-LATP particles in different temperature ranges. (**f**) Ionic conductivities of LATP, 1B-LATP, 1BP-LATP, and 1P-LATP pellets at room temperature (reprinted with permission from [96]; copyright 2022, Ceramics International).

Davaasuren et al. [95] studied the effects of the sintering aid Li_2_B_4_O_7_ on the microstructure and ionic conductivity of LATP. As the sintering temperature increases, the diffusion rate increases in favor of grain growth. The secondary phases of Li_3_PO_4_ and AlPO_4_ react with each other, thus making the LATP ceramics denser, inhibiting the growth of dendrites and reducing the grain boundary resistance, thus improving the ionic conductivity. The relative density reaches its maximum when the sintering temperature rises to 900 °C at a Li_2_B_4_O_7_ content of 0.5 wt% (Figure 9b).

Shen et al. [96] investigated the effects of sintering aids (Li_3_PO_4_, LiBO_2_·0.3H_2_O) and blends (0.32Li_3_PO_4_-0.68LiBO_2_·0.3H_2_O) on the sintering temperature and ionic conductivity of LATP electrolytes. It found that both reduced the sintering temperature of LATP to varying degrees and all increased ions (Figure 9e,f), but Li_3_PO_4_ was more significant, with a high ionic conductivity of 5.2 × 10^−4^ S cm^−1^ for LATP at 800 °C, 7.3 times that of pure LATP microspheres (7.1 × 10^−5^ S cm^−1^) sintered at 950 °C.

Kwatek et al. [97] investigated the effect of the sintering aid LiF on the sintering temperature and ionic conductivity of LATP electrolytes. It was found that the addition of LiF caused the decomposition of the resistive phase at the grain boundaries during the sintering process of LATP ceramic sheets, which reduced the interfacial impedance and thus increased the ionic conductivity. The highest ionic conductivity (1.1 × 10^−4^ S cm^−1^) at room temperature was achieved when the LiF molar ratio was 10% and the LATP ceramic electrolyte was sintered at 800 °C.

### 3.2. Cold Sintering Process

High temperature sintering leads to lithium loss and two-phase formation, so it is important to reduce the sintering temperature. Cold sintering process has recently been adopted, which can increase the density of ceramics at low temperatures while giving a compact structure [98].

Hamao et al. [91] conducted a two-step cold sintering process and investigated the effects of water demand. As shown in Figure 10a, good wetting and porous properties of LATP plates were observed at 650 °C. When the sintering time was 30 min, the pores in the LATP sheet were almost filled, forming a dense surface that inhibited dendrites growth (Figure 10b), reducing the interfacial impedance, and increasing the ionic conductivity (Figure 10c,d).

### 3.3. Interface Modification

The solid-solid contacts between electrodes and electrolyte in solid-state batteries is a difficult problem to overcome. It may induce side reactions and lithium dendrites, causing battery failure [100,101]. So, it is crucial to modify the contact interface. It is an effective way to obtain a denser and finer interface using nanoscale structural modification [102], which can improve the electrochemical stability of the electrode–electrolyte interface, meanwhile reducing the generation of lithium dendrites.

Wang et al. [99] set up a multifunctional SnO_2_ buffer (Figure 10e) on the surface of an LATP electrolyte using a drop coating method. The buffer creates a stable SEI layer which prevents the electrochemical reaction of LATP with lithium metal and reduces the corrosion of lithium anode. The buffer layer maintained close contact between electrodes during the volume expansion, reducing the interfacial impedance. The lithophilic properties of the layer can make the lithium deposit uniformly during the reaction and inhibit the growth of lithium dendrites, thus extending the battery life cycle and stability.

To improve interfacial properties, Stegmaier et al. [103] investigated interface engineering with a dopant of Mg^2+^. By analyzing doped LATP, they found that divalent Mg^2+^ as an interface doping can effectively improve the interfacial densification. In addition, as interphase cations replace each other, the contents decrease of Ti^4+^ and Al^3+^ lead to a decrease in electronic conductivity, protecting electrolytes from degradation. While, the increase in the content of Mg^2+^ and Li^+^ improve the lithium–ion conductivity. Therefore, interface doping provides a new direction for the study of solid electrolytes.

### 3.4. Block Copolymer

Diblock copolymer electrolytes are of interest because of their higher mechanical stiffness and ionic conductivity as well as better thermoplasticity compared to conventional solid polymer electrolytes. In addition, diblock copolymers consist of two different covalently anchored macromolecular chains in which the blocks can self-assemble into microphase-separated nanostructured forms such as spheres (SPH), hexagonal filled cylinders (HEX), bicontinuous gyroids (GYR), and lamellae (LAM). Nanostructured electrolytes made using self-assembly of diblock copolymers provide a range of independently regulated mechanical strength and electrochemical properties [104].

Huo et al. [105] used coarse-grained molecular dynamics simulations to discover the cascaded microphase structure of A_X_B_Y_-type diblock copolymers using the action of an applied electric field, and the permeable phase of charged blocks required for ion transport can be achieved at different block ratios using electric field adjustment. With increasing electric field strength, the copolymers with a block ratio of fA = X/(X + Y) = 0.67 undergo a laminar-columnar-disordered microphase transition; the copolymers with fA = 0.50 undergo a columnar-disordered microphase transition; and the copolymers with fA = 0.33 undergo a spherical-cylindrical-disordered transition (Figure 11). They also systematically investigated the formation mechanism and structural properties of each microphase, while summarizing the dependence of different morphologies of diblock copolymer electrolytes on electric field strength and orientation, block ratio, and system temperature; furthermore, it provides new directions for the design and development of new polymer electrolytes with pre-engineered structural/thermodynamic properties.

## 4. Conclusions

This review introduces the basic structure of LABs and the development of SSEs. As one of the next-generation energy storage devices, solid-state LABs have made many breakthroughs but not yet fully matured and still face many problems to be solved. The performance of LABs with different SSEs are summarized in Table 1. 

In recent years, many solutions have been proposed by research workers to address the shortcomings of solid-state electrolytes, but the solutions are not yet mature: (1) The advantage of adding sintering additives is that it reduces the sintering temperature of solid ceramic electrolyte, thus reducing the loss of lithium and the formation of the second term, which in turn improves the lithium ion conductivity, but whether the added sintering additives are compatible with the electrolyte material and whether the contact with the two poles is stable are potential problems. If the temperature is lowered more, the material will not easily produce the desired morphology, and if the temperature is lowered less, the loss of lithium and the formation of the second term will not be significantly reduced. (2) The cold sintering process is similar to the sintering aid process in that it lowers the sintering temperature, which reduces the loss of lithium and the formation of the second term, but it has a greater impact on whether the material can produce the desired morphology because it involves two sintering steps, and the more sintering steps, the more uncertainty. (3) The interface modification technique has the advantage of increasing solid–solid interface contact and decreasing interfacial resistance, but it does not address the loss of lithium and the formation of the second term during the high-temperature sintering of the solid ceramic electrolyte itself. (4) The diblock copolymer electrolytes offer independently regulated mechanical strength and electrochemical properties, which provide better mechanical strength and stability than conventional gel electrolytes, but still do not solve the problem of dendrite penetration compared to ceramic electrolytes. It is a complex and difficult task to investigate the actual material properties of the battery and to analyze the specific chemical changes during cycles, because the complex dynamic reaction process of the battery is differed from the structure, proportion, and fusion process of the specific actual material. It is necessary to combine the ex-situ and in-situ characterization techniques to conduct a comprehensive analysis of materials from the temporal and spatial dimensions.

It is a systematic work to consider the properties of each component in LABs. We need to consider the source of the key materials in the electrolyte, the cost of production, the difficulty of the manufacturing process, and whether they meet stability requirements. The price and catalytic effect should be considered in the selection of a cathode catalyst. Furthermore, the type of battery casing and the level of manufacturing process workmanship also influences the final performance of the battery, so it can be argued that any part of the battery determines the final performance of the battery. 

We firmly believe that through the continuous exploration of low-cost and highly Li^+^ conductivity stable SSEs, solid-state LABs will become the high energy density facilities for future mainstream application.

## Figures and Tables

**Figure 1 polymers-15-02469-f001:**
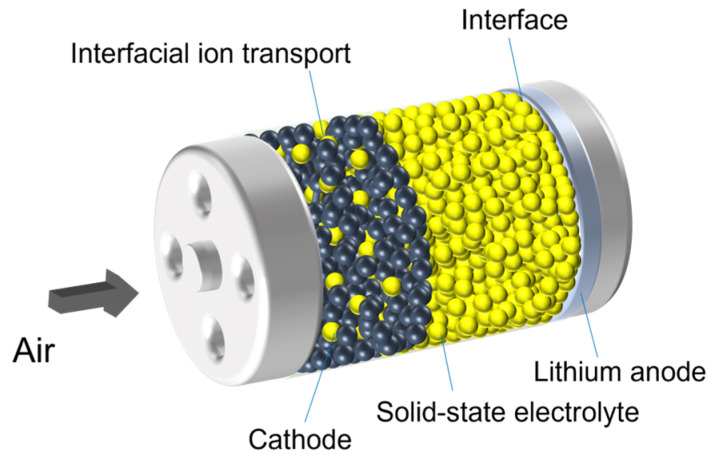
Schematic image of a solid-state Li–air battery.

**Figure 2 polymers-15-02469-f002:**
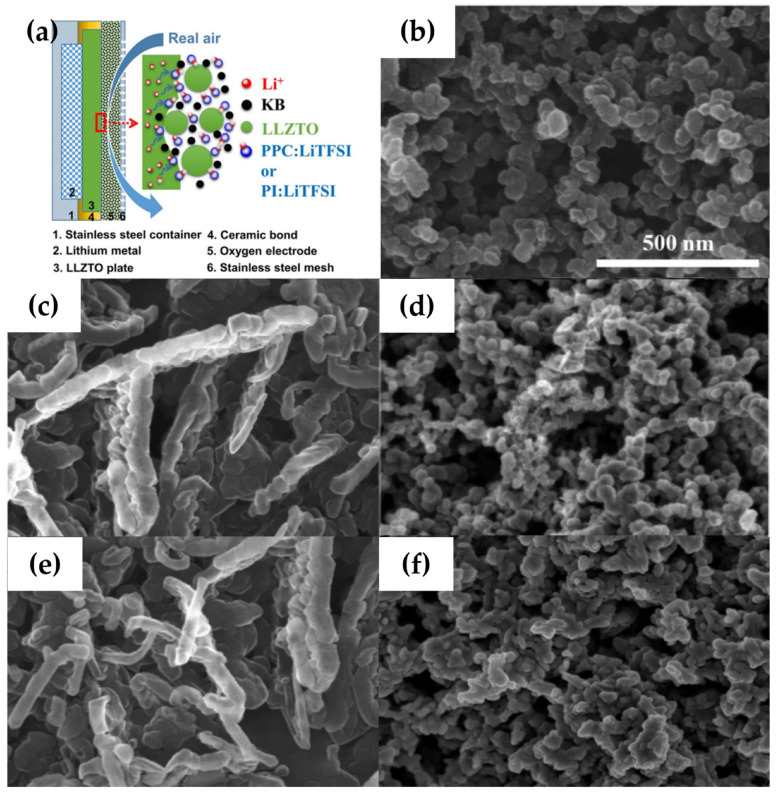
Schematic and morphological diagram of air cathode. (**a**) Schematic image. (**b**) Pristine air cathode. (**c**) First discharge to a capacity of 20,000 mAh g^−1^ carbon (~2.0 V). (**d**) First charge to 20,000 mAh g^−1^ capacity (~4.5 V). (**e**) Fifth discharge to 2.0 V, and (**f**) fifth charge to 4.5 V at 20 μA cm^−2^. White scale bars in all images represent 500 nm. (Reprinted with permission from [27]; copyright 2017, Scientific Reports).

**Figure 3 polymers-15-02469-f003:**
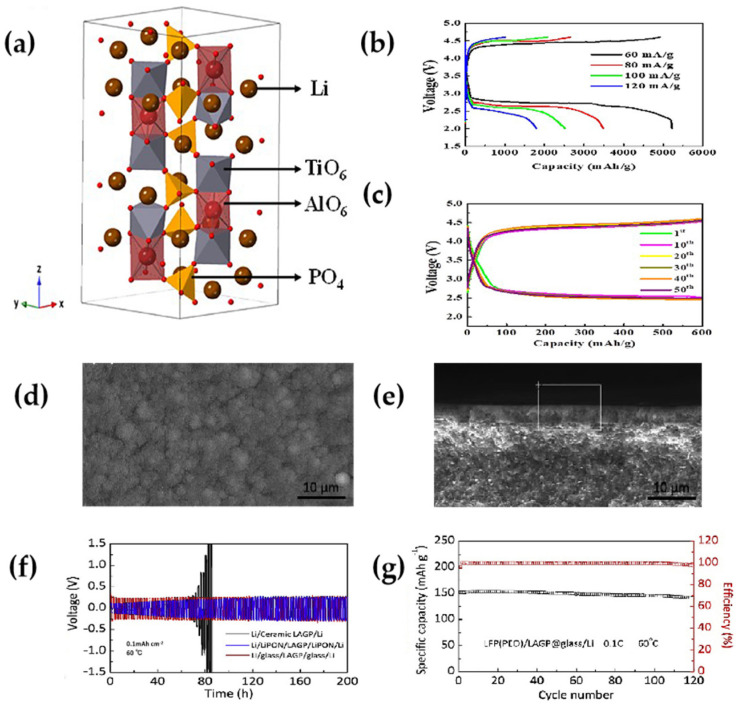
(**a**) Crystal structure of LATP. (**b**) Rate performance of Li–CO_2_ batteries at different current densities. (**c**) Discharge/charge curves of Li–CO_2_ batteries at a current density of 60 mA g^−1^ and a limited capacity of 600 mAh g^−1^ (reprinted with permission from [36]; copyright 2022, Electrochimica Acta). (**d**,**e**) LAGP@Glass surface, cross-sectional FESEM and EDXS. (**f**) Voltage profile versus cycle time for a Li/electrolyte/Li symmetric cell with a current density of 0.1 mA cm^−2^ at 60 °C. (**g**) Capacity retention LFP/LAGP@glass/Li at 0.1 C under 60 °C (reprinted with permission from [56]; copyright 2019, Electrochimica Acta).

**Figure 4 polymers-15-02469-f004:**
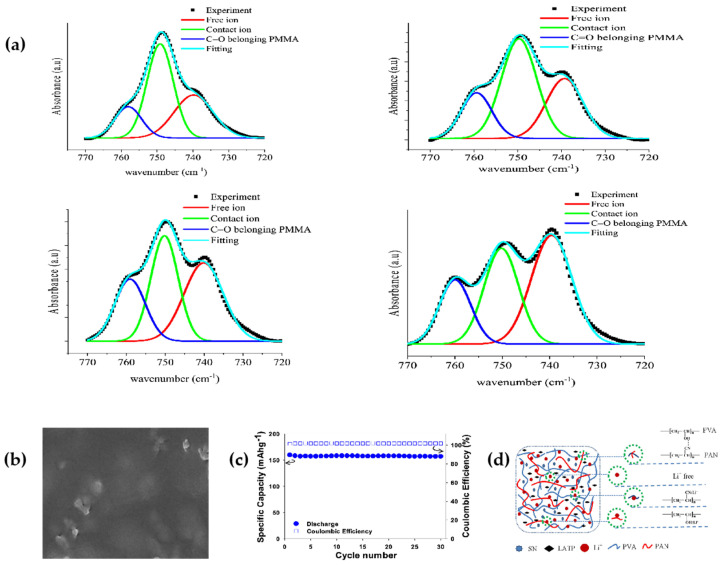
(**a**) FTIR deconvolution of PMMA-LiTFSI-TEGDME gel polymer electrolytes (reprinted with permission from [69]; copyright 2021, Optical Materials). (**b**) Top surface view of CPE film. (**c**) Cycling performance measured at 0.1 C. (**d**) Schematic diagram of the interactions among PVA, PAN, LiTFSI, LATP, and SN. (Reprinted with permission from [63]; copyright 2020, ACS Applied Energy Materials).

**Figure 5 polymers-15-02469-f005:**
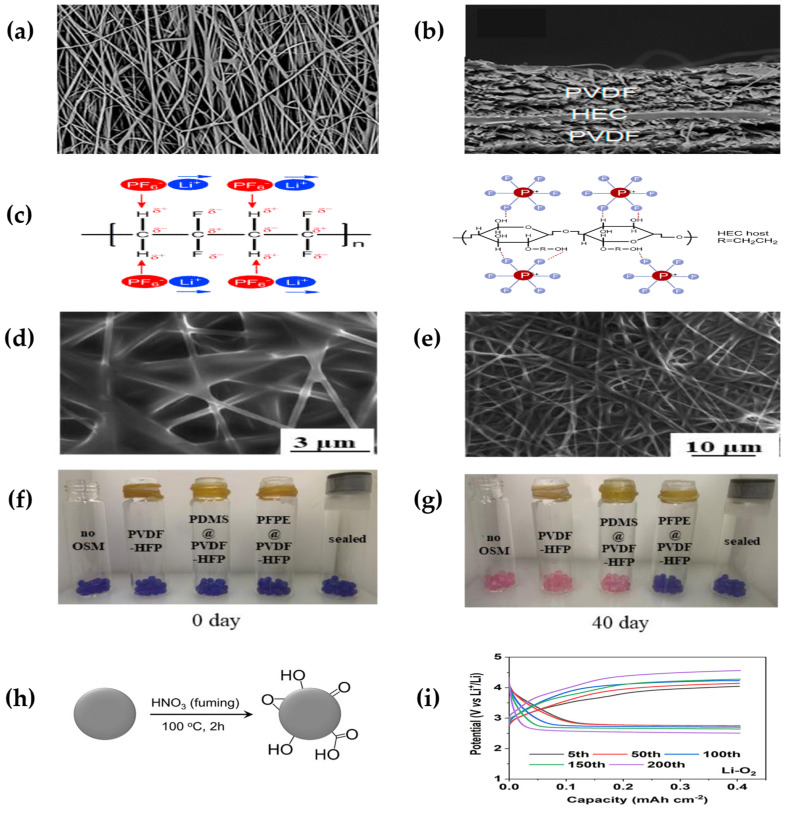
(**a**) Surface and (**b**) cross-section of PVDF/HEC/PVDF membrane. (**c**) Interaction between PVDF and LiPF_6_, lithium salt (LiPF_6_), and OH groups in HEC hosts. (Reprinted with permission from [74]; copyright 2017, Electrochimica Acta). (**d**,**e**) SEM optical photograph of PFPE@PVDF-HFP. (**f**,**g**) Photographs of silicone balls under different protection measures over time. From left to right refers to: without OSM; with protection of single PVDF-HFP; with protection of PDMS@PVDF-HFP; with protection of PFPE@PVDF-HFP; sealed treatment (reprinted with permission from [75]; copyright 2022, Electrochimica Acta). (**h**) Synthesis of OAC (black sphere animates active charcoal). (**i**) Galvanostatic cycling profiles Li−O_2_ battery with DPGE-OAC (reprinted with permission from [76]; Copyright 2021, ACS Catalysis).

**Figure 6 polymers-15-02469-f006:**
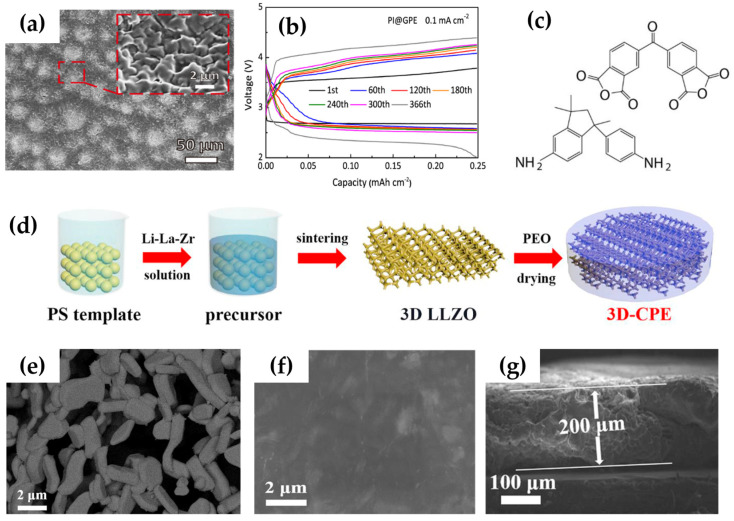
(**a**) XEM diagram of PI@GPE. (**b**) Discharge/charge curve of PI@GPE in lithium–oxygen battery. (**c**) Structural formula of polyimide membrane. (Reprinted with permission from [79]; copyright 2023, ACS Appl Mater Interfaces). (**d**) Schematic diagram of the preparation process of composite polymer electrolyte with 3D LLZO network. (**e**) 3D LLZO network grains surface (**f**) and cross-section of 3D-CPE under low. (**g**) Cross-section magnification of 3D-CPE (reprinted with permission from [80]; copyright 2020, Journal of Power Sources).

**Figure 7 polymers-15-02469-f007:**
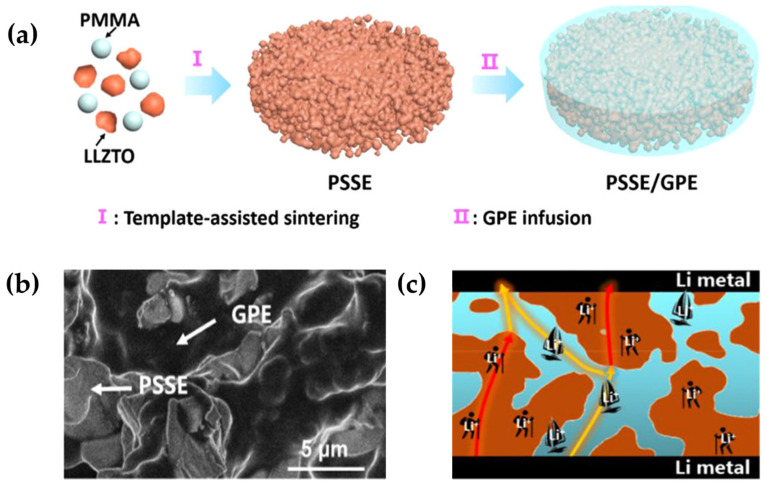
(**a**) Schematic diagram of PSSE/GPE synthesis. (**b**) Prepared SEM images of PSSE/GPE. (**c**) Schematic diagram of lithium–ion transport in PSSE/GPE (reprinted with permission from [81]; copyright 2020, Chemistry of Materials).

**Figure 8 polymers-15-02469-f008:**
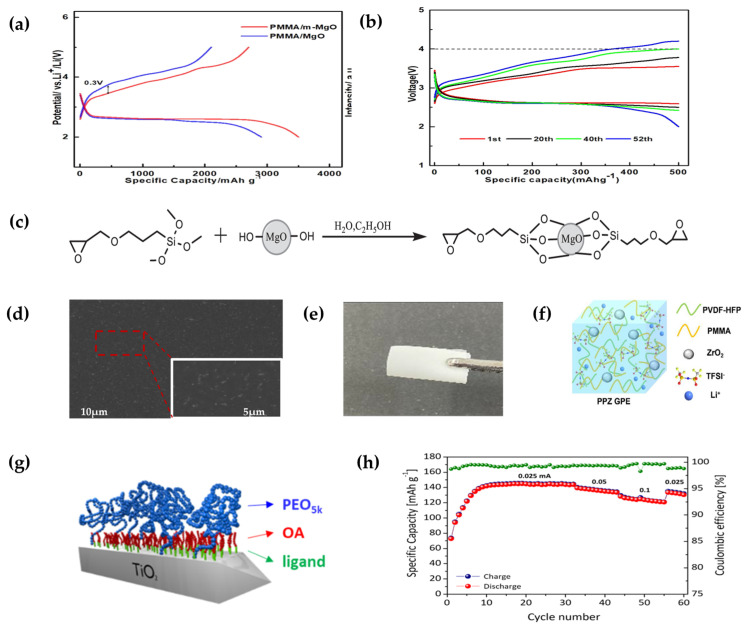
(**a**) Charge and discharge charge curves for a current density of 50 mA g^−1^ with PMMA/MgO electrolyte and PMMA/m-MgO electrolyte. (**b**) Discharge–charge curves of PMMA/m-MgO. (**c**) Preparation of the m-MgO particles (reprinted with permission from [87]; copyright 2021, Journal of The Electrochemical Society). (**d**) PPZ-6% films of SEM at different magnifications. (**e**) Optical photos in bending state. (**f**) Schematic illustrations of PVDF-HFP/PMMA-ZrO_2_ GPEs. (Reprinted with permission from [88]; copyright 2022, ACS Applied Energy Materials). (**g**) Ligand exchange reaction to obtain PEO5K@TiO_2_. (**h**) Specific capacity and coulombic efficiency (reprinted with permission from [89]; copyright 2020, Journal of The Electrochemical Society).

**Figure 10 polymers-15-02469-f010:**
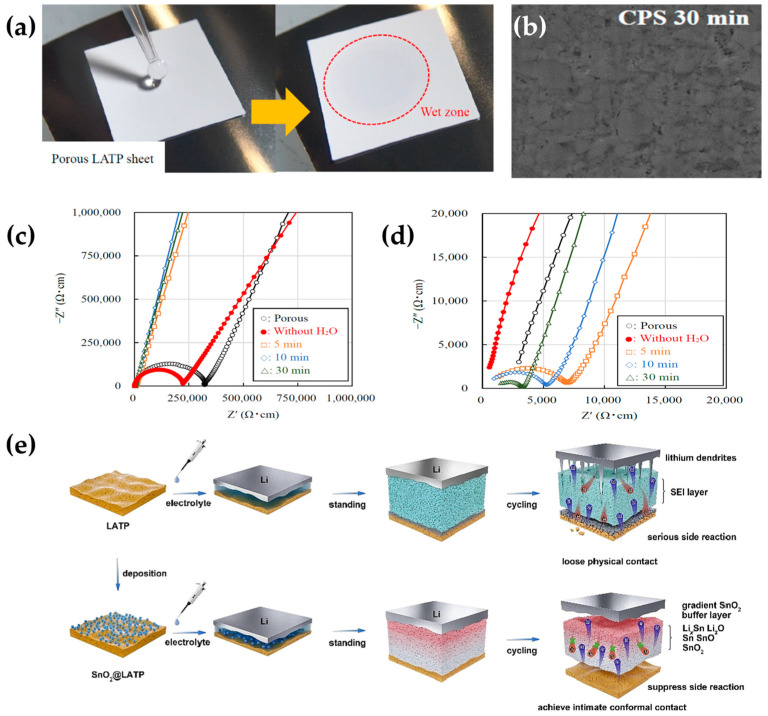
(**a**) Pictures of the produced LATP electrolyte tablets. (**b**) Cross-sectional SEM image of LATP electrolyte sheet. (**c**) Nyquist plots of LATP electrolyte sheets calcined at various conditions (**d**) Zoomed in the Nyquist plots. (Reprinted with permission from [91]; copyright 2021, Materials (Basel)). (**e**) Schematic representation of effective suppression of interfacial side reactions and improvement of loose physical contacts by constructing SnO_2_ GBL (reprinted with permission from [99]; copyright 2023, Journal of Energy Chemistry).

**Figure 11 polymers-15-02469-f011:**
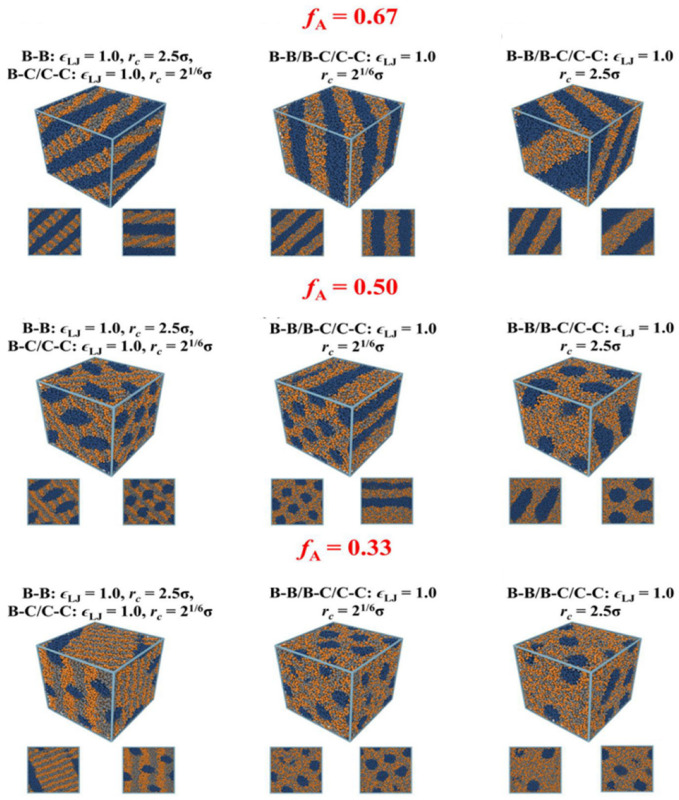
Representative snapshots of the simulations for diblock copolyelectrolytes at different block ratios, *f*_A_ = 0.67, *f*_A_ = 0.50, and *f*_A_ = 0.33 and at various interactions between charged beads (reprinted with permission from [105]; copyright 2023, Macromolecules).

**Table 1 polymers-15-02469-t001:** The performance of LABs with different SSEs.

Composites	Initial Capacity (mAh g^−1^) (Current Density)	End of Capacity (mAhg^−1^) (Cycle Number)	Ionic Conductivity (S cm^−1^) /Temperature	Type	Reference
LLZTO/PPC/LiTFSI	20,300/20 μA cm^−2^	-/(50)	1.6 × 10^−3^/RT	ISE	[27]
ZSM-5	-/400 mA g^−1^	70/480	-	ISE	[53]
UFSLAGP	152.2/(0.1 C)	-/31	1.6 × 10^−3^/RT	ISE	[60]
LAGP@glass	414/0.1 A g^−1^	142.5/120	9.85 × 10^−4^/60 °C	ISE	[56]
S-CPE	123/-	404.89/100	-	CPE	[68]
PMMA-LiTFSI	159.6/(0.1 C)	-/5	2.80 × 10^−4^/RT	GPE	[69]
PVAN50−20%LATP−10%SN	140/34 mA g^−1^	156.9/30	1.13 × 10^−4^/RT	CPE	[63]
PVDF/HEC/PVDF	6019/100 mA g^−1^	125/140	0.88 × 10^−4^/RT	GPE	[74]
PFPE@PVDF-HFP	-/0.1 mA cm^−2^	-/1200 h	-	SPE	[75]
PI@GPE	2485/0.05 mA cm^−2^	-/366	0.44 × 10^−4^/RT	CPE	[79]
3D-CPE	7540/312.5 mA g^−1^	1786/3	9.2 × 10^−5^/RT	CPE	[80]
PSSE/GPE	173/24 mA g^−1^	-/194	1.06 × 10^−3^/RT	CPE	[81]
LiBH_4_-MgO	35,111/50 mA g^−1^	162/5	2.86 × 10^−4^/RT	ISE	[86]
PMMA/m-MgO	153.0/0.5 C	-/52	7.76 × 10^−4^/RT	CPE	[87]
PVDF-HFP/PMMA ZrO_2_-6% (PPZ-6%)	120/0.1 A g^−1^	151.0/200	1.46 × 10^−3^/RT	CPE	[88]
PEO-TiO_2_ SnO_2_@LATP	157.6/0.1 C -	119/50 142.1/200	3 × 10^−4^/70 °C -	CPE ISE	[89] [99]

## Data Availability

Not applicable.

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
