# Peer review of "Solid-State Electrolyte for Lithium-Air Batteries: A Review"

_polymers, 2023, doi:10.3390/polym15112469_

Round 1

Reviewer 1 Report

The authors have done an exhaustive review of Lithium-Air Batteries. As conclusions, they should establish a comparison of the solutions studied, indicating the pros and cons of each solution presented.

In addition,

The review has been shown to be clear, complete. Being complete, it is relevant to the field of batteries. It is a very fascinating starting point for the work of researchers.

I am not aware of a similar review, so the current one is considered relevant to this field of research.

Statements and conclusions are consistent and supported by citations.

The figures/tables/images/schemes are appropriate and correctly show the data.

Author Response

Thank you for your comments.

Reviewer 2 Report

Qiancheng Zhu et al. reviewed Solid-State Electrolyte for Lithium–Air Batteries: A Review. The review was mainly focused on the solid state electrolyte for lithium battery. The authors has described the challenges of solid state electrolytes. The topic is highly exciting and the study is very informative.    

However, it requires minor revisions in order to meet the journal's requirements.

1. The authors can present the data more systematic way (I feel a table with literature will be interesting). The results can depicted in way that reader can corelate how researchers are succeeding to overcome the challenges. 

2. How solid state electrolyte science is evolving with time in battery is missing. 

3. Very recently, researchers are working on di block ionic polymers as a solid state electrolyte in battery. The authors has not touched this part. I feel it will be interesting if authors can consider to add block copolymers part in this review. 

 4. It will be easy for the readers to understand the science if authors present the chemical structure of the polymers.  

Author Response

Comment 1:  The authors can present the data more systematic way (I feel a table with literature will be interesting). The results can depicted in way that reader can corelate how researchers are succeeding to overcome the challenges. 

Response: Thank you for your advice. We added Table 1 depicted the performance of LABs with different SSEs in the conclusion part.

Comment 2:  How solid state electrolyte science is evolving with time in battery is missing. 

Response: We added the development of solid state electrolyte science in the Introduction. As follows:

The first generation of lithium ion solid-state electrolytes dates back to the 1830s when Faraday discovered that heated Ag2S and PbF2 had anionic conduction properties, but they did not develop rapidly because of their low ionic conductivity at room temperature, high interfacial impedance, and susceptibility to short circuit due to dendrite penetration [1]. It was not until the 1960s that β-Al2O3 was discovered to have two-dimensional sodium ion conduction properties and was subsequently applied to high-temperature sodium-sulfur batteries [2, 3]. Therefore, the 1960s is considered the starting point of the development of solid-state electrolytes, and solid-state electrolytes were first applied to batteries. In the following decade, Ag3SI solid-state lithium-ion conductor materials were successfully used for energy storage, solid-state electrolytes continued to be used in practice [4], and in 1973, the PEO polymer was discovered to have the ability to conduct lithium ions, thus expand the scope of solid-state ionics from inorganic materials to polymers [5], and since then, lithium-ion polymer conductors have emerged and been used in all-solid-state polymer lithium-ion batteries. In 1992, Oak Ridge National Laboratory successfully prepared LiPON thin film electrolyte material, which played a key role in improving the performance of thin film lithium batteries [6]. Since then, many types of inorganic solid-state electrolyte materials have emerged, including chalcogenide, sodium supersonic conductor (NASICON), garnet, sulfide, etc [7-10]. Until the early 21st century, solid-state electrolytes began to be combined with gaseous and liquid cathodes for lithium-ion batteries, such as solid-state lithium-air batteries, lithium-sulfur batteries, lithium-bromine batteries, etc [11-13].

Comment 3: Very recently, researchers are working on di block ionic polymers as a solid state electrolyte in battery. The authors has not touched this part. I feel it will be interesting if authors can consider to add block copolymers part in this review. 

Response: We have added the development of di block ionic polymers as a solid state electrolyte in battery. As follows:

Diblock copolymer electrolytes are of interest because of their higher mechan ical stiffness and ionic conductivity as well as better thermoplasticity compared to conventional solid polymer electrolytes. In addition, diblock copolymers consist of two different covalently anchored macromolecular chains in which the blocks can self-assemble into microphase-separated nanostructured morphologies such as spheres (SPH), hexagonal filled cylinders (HEX), bicontinuous gyroids (GYR), and lamellae (LAM). Nanostructured electrolytes made by self-assembly of diblock copolymers provide a range of independently regulated mechanical strength and electrochemical properties [104].

 Huo et al. [105] used coarse-grained molecular dynamics simulations to discover the cascaded microphase structure of AXBY-type diblock copolymers by the action of an applied electric field, and the permeable phase of charged blocks required for ion transport can be achieved at different block ratios by electric field adjustment. With increasing electric field strength, the copolymers with a block ratio of fA = X/(X + Y) = 0.67 undergo a laminar-columnar-disordered microphase transition; the copolymers with fA = 0.50 undergo a columnar-disordered microphase transition; and the copolymers with fA = 0.33 undergo a spherical-cylindrical-disordered transition (Figure 11). They also systematically investigated the formation mechanism and structural properties of each microphase, while summarized the dependence of different morphologies of diblock copolymer electrolytes on electric field strength and orientation, block ratio, and system temperature, besides, provided insights for the design and development of novel polymer electrolytes with pre-designed structural/thermodynamic properties.

 Comment 4: It will be easy for the readers to understand the science if authors present the chemical structure of the polymers.  

Response: Thank you for your advice. Chemical structures of the polymers are now given in Fig 4, Fig 5, Fig 6 and Fig 8.
